# The Impact of a Change in Employment on Three Work-Related Diseases: A Retrospective Longitudinal Study of 10,530 Belgian Employees

**DOI:** 10.3390/ijerph17207477

**Published:** 2020-10-14

**Authors:** Laura Maniscalco, Martijn Schouteden, Jan Boon, Domenica Matranga, Lode Godderis

**Affiliations:** 1Department of Biomedicine, Neuroscience and Advanced Diagnostics (BIND), University of Palermo, 90127 Palermo, Italy; 2IDEWE, External Service for Prevention and Protection at Work, 3001 Heverlee, Belgium; Martijn.Schouteden@idewe.be (M.S.); Jan.Boon@idewe.be (J.B.); lode.godderis@kuleuven.be (L.G.); 3Department of Health Promotion, Maternal-Child Health, Internal and Specialized Medicine of excellence “G. D’Alessandro”, University of Palermo, 90127 Palermo, Italy; domenica.matranga@unipa.it; 4KU Leuven, Centre for Environment and Health, 3000 Leuven, Belgium

**Keywords:** chronic diseases, cardiovascular diseases, musculoskeletal diseases, neuropsychological diseases, work-related risks

## Abstract

Background: The literature that has investigated to what extent a change in employment contributes to good health is contradictory or shows inconsistent results. The aim of this study was to investigate whether an association exists between a change in employment and cardiovascular, musculoskeletal and neuropsychological diseases in a sample of 10,530 Belgian workers in a seven-year follow-up study period. Methods: The following factors were analysed: Demographic variables, a change in employment and the work-related risks. Individuals being on medication for cardiovascular, musculoskeletal, and neuropsychological diseases were used as proxies for the three health issues. Logistic regression models for autocorrelated data with repeated measures were used to examine each medication type. Results: A change in employment and psychosocial load can have an important effect on the health of cardiovascular employees. Demographic variables, such as BMI and age, are risk factors for all three medications. Repetitive, manual tasks, handling static, exposure to noise levels of 87 dB, mechanical and/or manual handling with loads, and shift work were found to be positively associated with medications taken for musculoskeletal diseases. Exposure to noise 80 dB(A), managing physical loads and night work were found to be associated with being on medication for neuropsychological diseases. Physical activity and skill levels were considered to be protective factors for being on medication for neuropsychological diseases. Conclusions: Change in employment and psychosocial load were found as two important risk factors for being on medication for cardiovascular (CVD). Dealing with loads, doing shift work and being daily exposed to the noise of 87 dB correlated with being on medication for musculoskeletal (MSD). Dealing with physical loads, doing night work and being exposed to the noise of 80 dB were risk factors for being on medication for neuropsychological (NPD). While doing physical activity and reporting higher skill levels were found to be protective factors for NPD.

## 1. Introduction

It is reported that on average people change jobs twelve times during their career. A longitudinal study published by the Bureau of Labour Statistics demonstrated that men had a total of 12.1 jobs and women a total of 11.6 jobs between the ages of 18 and 46. Typically, the main reasons for changing employment are related to: The desire to increase one’s pay or to improve one’s work-life balance, career advancement, the choice of a less stressful job, incompatibility problems with the boss, and so on [1]. On the contrary, job loss can have a detrimental effect on health as it is negatively associated with: a higher rate of mortality, poor general health, long-standing illness, poorer mental health, psychological distress, and higher hospital admission rates [2,3]. Health and employment influence each other in a reciprocal relationship. Health has a direct influence on work, thus healthy people are more likely to obtain and remain in employment. Conversely, individuals with existing health issues are more likely to be hired for a job with poor working conditions, which in turn can worsen pre-existing health conditions [4,5]. Health also plays an important role in work capabilities, which is influenced by physical and psychosocial demands at work, and by the employees’ mental and physical capabilities and lifestyle factors. Disequilibrium between these determinants in suffering ill health could have consequences on work performance, including productivity loss, sickness-related absence, and work-related disabilities [6]. The current literature demonstrates the conflicting results of the effects of employment on health with positive and negative effects [2,7]. One the one hand, employment has been considered as beneficial to health, especially for depression and general mental health because it improves physical and psychological well-being and it provides opportunities to: Increase one’s skills, social integration, life goals, prestige and purpose, thereby achieving a sense of personal achievement [2,8,9]. Work-related risks that may impact workers’ health can be grouped as biological, physical, ergonomic, chemical, and psycho-social. National Institute for Occupational Safety and Health (NIOSH) has reported that there exist 29 kinds of physical, 25 kinds of chemical, 24 kinds of biological, six kinds of ergonomic, and 10 kinds of psychosocial hazards (as work-stress) [10]. Indeed, work can have negative effects on health due to the exposure to health-harming physical and psychosocial stressors, such as: demanding physical work, exposure to various types of harmful radiation, excessive vibration, and high levels of noise and polluted air in the workplace [8,11]. Moreover, the nature and quality of work should be taken into account in exploring the relationship with health since the influence of work on health is positive providing that working conditions are favourable; conversely, if employment conditions are poor, then work can impair health [7,8].

Other noteworthy factors are related to changes in employment status and working conditions, which can be repeated throughout one’s career. Consequently, the question arises: To what extent does a change in employment contribute to good health? In a sample of Swedish male workers, poor mental health has been found to be weakly associated with the frequency of a change in employment, adjusted for socio-economic status and mediated by general mental health [12]. Another study, in a sample of male employees, found the highest mortality risk in association with a series of changes among unrelated jobs. This sample was retrieved from the Stanford-Terman longitudinal study, an archive containing detailed work and life histories on approximately 1500 men and women. Education, occupation, physical health, anxiety and depression were found as significant risk factors [13]. In a sample of young Dutch subjects, voluntary changes in employment was associated with better health condition. The study included work perceptions (organizational, departmental and task-related as explanatory variables) and job satisfaction, organizational commitment, intention to leave, absenteeism and tardiness as work outcomes [14]. Finally, in a cohort of members of the National Survey of Health and Development, a longitudinal study of people born throughout Britain, it was found that early job changing is an indicator of later psychiatric problems [15].

Conversely, other studies did not find any evidence between frequent job changes in employment and the worker’s health status. This result was found for cardiac health in a sample of Scottish employees [16] and myocardial infarction in a case-control study in Sweden [17].

Given the controversial nature of these wide-ranging outcome variables, the authors of this study contend that the possible association between frequent changes in employment and health merits further attention. The literature that has investigated to what extent a change in employment contributes to good health is contradictory or shows inconsistent results. For, this reason the aim of this study was to detect whether there exists an association between a change in employment and cardiovascular (CVD), musculoskeletal (MSD) and neuropsychological (NPD) diseases in worker’s employed in different sectors in a seven-year follow-up study period. The association between changes in employment and being on medication for the aforementioned work-related diseases will be scrutinised, as proxies for the three health problems, while controlling for work-related risks.

## 2. Materials and Methods 

### 2.1. Population and Study Design

The study data were obtained from the largest central repository of data on Belgian employees, the IDEWE data warehouse. IDEWE is the Belgian *External Service for Prevention and Protection at Work*. IDEWE owns an administrative database, including data from the annual health checks of Belgian employees, and recorded and encoded these data in an electronic format using international or national classification standards [18]. Detailed information about the data collection and data warehouse have been described earlier [18,19]. Periodic health checks in Belgium are mandatory for employees, who are exposed to occupational hazards [20]. In addition to medical data, personal and work characteristics are also registered and encoded during medical examination, using international or national classifications standards in electronic records. Thereafter, the data stored in electronic medical files were extracted, translated and loaded into a data warehouse for further analysis. The study protocol received the ethical approval from the Ethical Committee of Azienda Ospedaliera Policlinico “Paolo Giaccone” of Palermo (Nr. 8/2020, dated 23 September, 2020).

### 2.2. Data Collection and Variables

The final dataset contained 73,710 observations (10,530 employees with at least seven measurement time points between 1993 and 2019) and 24 variables, after removing subjects lacking gender and skill levels. The covariates included in the analysis were: Demographic (age, sex), physical and behavioral characteristics (such as high blood pressure, overweight and obesity measured through BMI, smoking habits), occupational (such as a change in employment, skill levels and sector) and employment-related risk. The *medication use* response variable was a binary variable with the *Not a user* category (indicating that a subject in a particular year did not take any medication) and the *user* category. Since medication compliance is an accurately-encoded registration type in medical files, it was used as a proxy for health status. Limiting the discussion to CVD, it was assumed that an individual took lifelong medication. A “change in employment” is a binary variable with *yes* or *no* responses if the employee had not changed employment in that year compared to the previous year. Measured by considering the ISCO-08 code, indicating the work type associated with a particular occupation, skill levels were used as a proxy of educational level. Specifically, the first skill level corresponds to the lowest educational level (primary school), and the second skill level includes employment requiring a secondary education. Occupations classified in the third skill level require a high level of literacy, numeracy and well-developed interpersonal communication skills, and the fourth level requires very high skills, involving an undergraduate or Master’s degree and an advanced research qualification. The various employment sectors included 10 major groups: Education, Healthcare, Government, Accommodation and Food Services, Distribution trades, Manufacturing, Services, Construction, Transport and storage, and Other. These categories are based on the official statistical classification of economic activities in the European Community (the NACE code). According to Belgian legislation, occupational physicians perform an individual risk assessment based on information received from the employer and information given by each employee during the medical examination. The occupational physician encodes risk factors using an electronic record system, given a list of risk codes that are briefly described using an overview list. This list was designed to allow for the interpretation of the physician, based on his/her own experience and based on the needs of every specific employer or sector at hand. From this list of over 400 risk codes, a small subset was selected by two occupational physicians in an independent way, until consensus was reached. (See Appendix A). For this reason, some risks related to specific employment types were also considered. These were included as *yes/no* binary variable categories: stress at work, improper or unacceptable behavior at work, work-related exhaustion, noise at work (exposure values of 80 dB, exposure values of 85 dB and exposure values of 87 dB) [21], mechanically and/or manual handling with loads, manual lifting/holding/carrying, manual pulling and/ or pushing, manual repetitive tasks, handling static loads, shift work with and without task-specific risks, night work with and without task-specific risks, and a psychosocial load. The blood pressure was categorized according to the American Heart Association classification. This classification considers both systolic and diastolic blood pressure. The variable was categorized as “*Normal*”, “*Elevated*”, and “*High*”. *Elevated* for systolic blood pressure between 120–129 and diastolic blood pressure less than 80 and *High blood pressure* for a systolic pressure more or equal than 140 and a diastolic pressure more or equal than 90. Smoking habits were classified as “*No*” if the subject were not a user or a former user, and “*Yes*” if the subject was currently a smoker. 

### 2.3. Statistical Methods

Continuous variables were summarized as a mean value (SD = standard deviation), and categorical variables were analyzed as *counts* and *percentages*. In order to assess the statistical significance of this difference, the Chi-squared test was used for categorical variables (or Fisher exact test when necessary), and the t-test was used for continuous variables. In order to assess the relationship between a change in employment and medication use (while controlling for the time-variant and time-invariant confounders), a logistic regression model for autocorrelated data with repeated measures was used for each type of medication relating to cardiovascular, musculoskeletal and neuropsychological diseases [22]. For all categorical variables, the *no* category was used as reference, and separate models were stratified by gender. A backwards, stepwise selection process with the Akaike information criterion (AIC) for model selection was also deployed and the block bootstrap method was used to generate confidence intervals. For the sake of simplicity, in the multivariable analysis only coefficients with significant p-values were reported. The data were analyzed using R software, version 3.5.1 including the Bild packages (version 1.1). A *p*-value less than 0.05 was considered statistically significant. 

## 3. Results

### 3.1. Descriptive Statistics

Table 1 shows the results of the descriptive analyses. The sample included 4769 (45%) females and 5761 (55%) males. The mean age of females was higher compared to that of males (Female: 38.89 ± 9.36; Male: 37.56 ± 9.77), while BMI was on average higher for males (BMI: 26.08 ± 4.15) when compared to females (BMI: 25.27 ± 4.86). The blood pressure variable was distributed differently between males and females: at baseline 73% of males, compared to 52% of females, suffered from high blood pressure. Typically, both males and females were not smokers at baseline during the data collection period. The distribution by activity sector and gender revealed that 40.8% of males worked in the manufacturing sector, followed by 15% in the government, 10.7% in distribution trades and 10.3% of males worked in the healthcare sector. Whereas, the distribution of female employees by sector was the following: 74.6% worked for the healthcare sector, 9.2% for government and 6.6% in the manufacturing sector. The majority of males were characterized by the second skill level (74%), while most females reporting the third or fourth skill level (42%). At baseline few males and females were affected by stress at work and exhaustion due to employment-related risks. The majority of males were subjected to daily noise levels of 87 (dB) (51%), as were males and females regarding manual load-handling (59% for males, 88% for females). Considering that the study is a retrospective (historical) cohorts percentage of people exposed to daily noise levels of 87 dB could appear oversized. At univariable analysis, male employees changing employment versus those who did not show differences in: BMI, stress at work, work-related exhaustion, noise levels of 85 dB or 87 dB, mechanically and/or manual handling with loads, holding, carrying, handling static loads, shift work without task-specific risks and psychosocial loads. Similarly, female employees who changed employment versus those who did not displayed differences in: BMI, skill level, static load, shift work (with and without task-specific risks), night work (without task-specific risk) and psychosocial loads. At baseline, 2% of males were medication compliant for MSD, compared to 4% of females (*p*-value < 0.0001); 7% of males and 7% of females (*p*-value = 0.418) were medication compliant for CVD; 2% of males compared to 5% female users (*p*-value < 0.0001) (Table 2) were medication compliant for NPD. 

### 3.2. Being on Medication for MSD 

BMI, age and performing manual, repetitive tasks were found to be positively associated with being on medication for MSD for males and females. Indeed, an increase in these variables corresponded to a higher probability of medication use. A change in employment did not seem to be associated to MSD for males and females. In addition to the aforementioned MSD variables, the following increased the probability reporting MSD in males: Exposure to noise 87 dB level, dealing with physical load, manual handling with loads and shift work (OR = 1.22 95%CI [1.04–1.40], OR = 1.25 95%CI [1.05–1.47], OR = 1.21 95%CI [1.03–1.42] and OR = 1.70 95%CI [1.29–2.11] respectively). In contrast, doing physical activity and manual lifting reduced the probability to be on medication for MSD (OR = 0.85 95%CI [0.75–0.95], and 0.33 95%CI [0.22–0.38] respectively). The risk for females dealing with static loads at work increased the risk of being a medication user (OR = 1.62 95%CI [1.22–2.11]) whereas the manual dealing with loads was found to be a protective factor (OR = 0.79 95%CI [0.68–0.95]) (Table 3).

### 3.3. Being on Medication for CVD 

A change in employment seemed to have a significant effect on the response variable for both genders (OR = 1.93 95%CI [1.47–2.78] for males and OR = 1.86 95%CI [1.34–2.50] for females). This implies that the probability of those changing employment to be on medication for cardiovascular diseases was approximately 93% and 86% for males and females respectively, and this is higher than the odds for those who do not change employment after controlling for other explanatory variables. Moreover, BMI, age and psychosocial pressure were found to be significant for being on medication for CVD. Indeed, the psychosocial load seems to increase the probability of being on medication for CVD with OR = 1.23 (95%CI [1.07–1.38]) for males and OR = 1.21 (95%CI [1.09–1.32]) for females. In addition to the aforementioned variables, the following were positively associated with CVD in males: stress at work, noise levels of 80 dB (A), manual handling with loads in the workplace, dealing with static loads, shift work (without task-specific risks) and night work (with and without task-specific risks). Moreover, smoking played a confounding role as non-smokers in the study sample were over-represented with respect to smokers (2.5:1). However, the model for females showed that dealing with physical loads as positively associated with the response variable (Table 4). 

### 3.4. Being on Medication for NPD 

To be on medication for NPD was significantly associated with age (OR = 1.02 95%CI [1.01–1.03] for male, OR = 1.01 95%CI [1.01–1.02] for female), negatively associated with physical activity (OR = 0.83 95%CI [0.74–0.94] for male, OR = 0.86 95%CI [0.78–0.96] for female). Furthermore, males with skill level 3 or 4 were protected compared to skill level 1 (OR = 0.63 95%CI [0.40–0.99]) and both males and females with skill level 2 were protected, compared to individuals with a skill level 1 (OR = 0.58 95%CI [0.42–0.89] for male, OR = 0.77 95%CI [0.60–0.99] for female). Moreover, BMI, noise levels of 80 dB(A), night work (without risks) were also associated with being on medication for NPD in the male sample. Specifically, noise levels of 80 dB(A) and night work (without task-specific risk) increased the probability of being on medication for NPD (OR = 1.58 95%CI [1.20–2.03] and OR = 1.34 95%CI [1.01–1.79] respectively). However, an increase in BMI decreased the probability of reporting NPD. The following was found to increase the probability of being on medication for NPD in females: History of smoking, dealing with physical loads in the workplace and doing night shifts (with task-specific risk). Furthermore, female employees in the healthcare and government sectors presented higher risks associated with being on medication for NPD, compared to female employees in the education sector (OR = 2.20 95%CI [1.24–4.31], and 2.35 95%CI [1.20–5.08], respectively) (Table 5).

## 4. Discussion

The aim of this work was to assess whether an association exists between a change in employment and the use of medication for three work-related diseases: Cardiovascular, musculoskeletal and neuropsychological. These subjects were then followed up in a seven-year period, while they were in employment. To this aim, a large data warehouse obtained from IDEWE, a Belgian *External Service for Prevention and Protection at Work*, was deployed. The application of three logistic regression models for autocorrelated data revealed important results. 

Exposure to risk factors can lead to MSD over extended periods of time but it is highly improbable that the results of a change in employment would have an immediate effect on MSD. Due to their non-specific nature, MSDs are often significantly under-reported in the literature. Moreover, it has been documented that some social security records are subject to a prolonged time delay between the initial declaration and recognition of MSD [23]. This study is for MSDs in line with the literature. Individual factors such as BMI, age and physical activity are associated with MSDs [23] and other work-related risk factors include: Unnatural posture, repetitive strain injury, physical exertion, static work; being subject to excessive vibrations, work overload, stress and other psychosocial factors can all contribute to the onset of those disorders [24]. The sectors mentioned in this study were not statistically significant for MSDs and it is posited that this may be due to the nature of the sample composition. Indeed, IDEWE mainly collects data from employees who work in the healthcare sector. Furthermore, it has been demonstrated elsewhere that agriculture, health and social work, transport storage and communication sectors were more frequently associated with MSD [23]. Lifting and especially lifting of heavy loads, sedentary work and physical inactivity have been found to contribute to back pain. The protective role of manual handling with loads for being on medication for MSD may well be due to “healthy worker effects” and uncontrolled confounding. Many authors believe that work strain mediates the association between work stressors and work-related musculoskeletal complaints, whereby the mental and physical mechanisms involved elicit muscle tension and induce musculoskeletal pain [25].

An important issue regarding being on medication for CVD is due to the deteriorating role of changes in employment, as confirmed by Haynes S. [26]. This is due to the fact that individuals who experienced frequent changes in employment are more likely to smoke, to consume excess alcohol, and to do less physical activity [15]. Moreover, this study found that increasing age is a significant risk factor for CVD, as also confirmed by the WHO [27]. Furthermore, it has been asserted that a higher skill level reduces the probability of being on medication for CVD, even if not significant in this study. In line with the results of the Belgian Job Stress Project [28], psychosocial pressure was found a determinant of being on medication for CVD. Indeed, the WHO have found the following to be associated with an increased risk of being on medication CVD: Mental pressure at work, psychosocial stress, sedentary work, chronic exposure to excessive levels of noise and other occupational factors [27,29]. Our study found that physical load, manual handling with loads and static load are risk factors for CVD. This is in accordance with Hannerz and Holtermann’s study where employment in occupations that involve heavy lifting is a predictor of Ischemic Heart Disease [30]. Other studies also reported a positive relationship between shiftwork and coronary heart disease, possibly due to irregular working hours or unbalanced lifestyle [27]. 

The lack of association between a change in employment and being on medication for NPD could be due to the social stigma and consequent under-reporting of disease [31]. This implies a structural discrimination in workplace settings since individuals with mental illness tend to have reduced access to quality jobs and they are less likely to be perceived as being suitable for promotion [32]. While, it is likely that people with mental health issues may change their employment, in order to reduce workload, stress and responsibility, the opposite is not necessarily true. It is unlikely that a change in employment may be associated with employees recovering from mental issues since these are often chronic in nature [33]. Undertaking physical activity and higher skill levels were found to be protective factors [34], while age is a risk factors for mental health in this study and this is in line with the findings of the WHO. The presence of respiratory system complaints caused by smoking habits was found to be an important risk factor for a broad category of mental disease as anxiety, depression, and the combination of anxiety and depression [35]. 

The study found the following risk factors for being on medication for NPD: dealing with physical loads and night work (with task-specific risks) for women, and night work (without task-specific risks) and excessive daily noise at work for men. It seems that psychological ill health is worse for female employees in the healthcare and government sectors as compared to female non-healthcare employees (for example, those working in the education sector). This is in accordance with Stansfeld et al. that demonstrated that the type of occupation is an important risk factor for common mental disorders. It occurs because “occupations may be typified by high levels of job demands, especially emotional demands and lack of job security. The reasons why occupations have low rates of common mental disorder are varied and may include high levels of job discretion, good job training and clearly defined job tasks” [36]. An intervention study, which has been included in the literature review, has demonstrated that aerobic exercise is associated with improved health [37], and a Canadian population-based longitudinal study concluded that work-related stress is the major risk of depressive episodes [38]. The same Canadian authors also stated that the impact of risk factors may vary across genders since the impact of work stressors on common mental disorders in their review differed for males and females [38]. Mental health is demonstrated to be negatively correlated with the constant high physical and psychosocial work demands [39]. A study conducted by the WHO regarding healthcare employees showed that an excessive workload is a risk factor for mental health, and the authors of this paper observed that smoking can increase neuropsychological medication use in women [40].

The main strength of the paper is the availability of a seven-year follow-up study period, in order to investigate the association between job changes and three different diseases, while considering a wealth of work-related risk factors. Conversely, there are a few study limitations. First of all, the drop-out of employees that leave their job or change it, with the effect to be lost to follow up, because they are no longer enrolled in the same OSH provider (IDEWE). Moreover, the specific causes of job changes are not considered, so the occurrence of some confounding in the analysis cannot be excluded. Other important potential risk factors such as diet, work satisfaction, and sickness absence, family life, supportive relationships with colleagues, economic security, and access to social support were not measured. Self-reported information on smoking habits was potentially underreported and some risks as burnout, conflicts with customers, safety risk and hindrance were excluded to avoid sparseness in the models, since they presented very few observations.

## 5. Conclusions

The aim of this study was to explore the effect of changes in employment on the health of employees, while considering a series of risk factors associated with work-related diseases. This association was found to be statistically significant and positive only for CVD, excluding the effect of other covariates. Moreover, psychosocial loads also play an important role in the onset of CVD. Regarding medication for MSD, a positive association was found with BMI, age, manual and repetitive tasks, the handling of static loads, noise exposure of 87 dB, mechanical and/or manual handling with loads, and shift work. Finally, being on medication for NPD showed significant positive association with age, BMI, smoking habits, noise of 80 dB(A), dealing with physical loads and night work (without task-specific risk), while doing physical activity and reporting higher skill levels were found to be protective factors. As a consequence, it is recommended that the employer’s working life is tracked by recording his/her job changes. In fact, the worsening of the worker’s health reflects in lower productivity, augmented cost for the employer and affects public health. Our study showed an increased risk of cardiovascular disease in Belgian workers who experience a job change. Therefore, it is advisable, especially in the transition phases between one job and another, that the worker’s health status is monitored by the general practitioner and the occupational physician.

## Figures and Tables

**Table 1 ijerph-17-07477-t001:** Baseline characteristics 10530 Belgian workers by gender.

Variables	Male	Female	*p*-Value
Age, mean (SD)	37.56 (9.77)	38.89 (9.36)	<0.0001
BMI, mean (SD)	26.08 (4.15)	25.27 (4.86)	<0.0001
Blood pressure			
Normal	682 (12%)	1456 (31%)	
Elevated	879 (15%)	856 (18%)	<0.0001
High	4200 (73%)	2457 (52%)	
Smoking habits			
No	3784 (66%)	3735 (78%)	<0.0001
Yes	1977 (34%)	1034 (22%)	
Sector			
Education	74 (1.3%)	144 (3.0%)	<0.0001
Healthcare	592 (10.3%)	3558 (74.6%)	
Government	862 (15.0%)	437 (9.2%)	
Food	23 (0.4%)	47 (1.0%)	
Distributive trade	618 (10.7%)	138 (2.9%)	
Manufacturing	2349 (40.8%)	316 (6.6%)	
Services	162 (2.8%)	54 (1.1%)	
Construction	542 (9.4%)	1 (0.0%)	
Transport	209 (3.6%)	17 (0.4%)	
Other	330 (5.7%)	57 (1.2%)	
Skill levels			
1	448 (8%)	1146 (24%)	<0.0001
2	4281 (74%)	1627 (34%)	
3 or 4	1032 (18%)	1996 (42%)	
Risk: stress at work			
No	5759 (99.9%)	4768 (99.9%)	1
Yes	2 (0.01%)	1 (0.01%)	
Risk: burnout			
No	5761 (100%)	4769 (100%)	1
Yes	0 (0%)	0 (0%)	
Risk: noise 87 (dB)			
No	2813 (49%)	4534 (95%)	<0.0001
Yes	2948 (51%)	235 (5%)	
Risk: noise 85 (dB)			
No	5451 (95%)	4743 (99%)	<0.0001
Yes	310 (5%)	26 (1%)	
Risk: noise 80 (dB)			
No	5661 (98%)	4749 (99%)	<0.0001
Yes	100 (2%)	20 (1%)	
Risk: Mechanical handling with loads			
No	5457 (95%)	4636 (97%)	<0.0001
Yes	304 (5%)	133 (3%)	
Risk: Manual handling with loads			
No	2338 (41%)	560 (12%)	<0.0001
Yes	3423 (59%)	4209 (88%)	
Risk: Manual lifting, holding, carrying			
No	5761 (100%)	4769 (100%)	1
Yes	0 (0%)	0 (0%)	
Risk: Manual pulling and pushing			
No	5761 (100%)	4769 (100%)	1
Yes	0 (0%)	0 (0%)	
Risk: Manual repetitive tasks			
No	5761 (100%)	4769 (99.9%)	1
Yes	0 (0%)	0 (0.01%)	
Risk: Handling static load			
No	5696 (99%)	4750 (99%)	<0.0001
Yes	65 (1%)	19 (1%)	
Risk: Shift work without task-specific risk			
No	4995 (87%)	4317 (91%)	<0.0001
Yes	766 (13%)	452 (9%)	
Risk: Shift work with task-specific risk			
No	5498 (95%)	4645 (97%)	<0.0001
Yes	263 (5%)	124 (3%)	
Risk: Night work without task-specific risk			
No	5515 (96%)	4585 (96%)	0.31
Yes	246 (4%)	184 (4%)	
Risk: Night work with task-specific risk			
No	5626 (98%)	4701 (99%)	<0.0001
Yes	135 (2%)	68 (1%)	
Risk: Psychosocial load			
No	5579 (97%)	4637 (97%)	0.263
Yes	182 (3%)	132 (3%)	

*p*-values from t-test for continuous variables and Chi-squared tests (or Fisher exact test when necessary) for categorical variables. Continuous variables are expressed as mean and SD and categorical variables as n and percentage. Elevated blood pressure: 120 ≤ Systolic ≤ 129 & Diastolic < 80; High blood pressure: Systolic ≥ 140 & Diastolic ≥ 90.

**Table 2 ijerph-17-07477-t002:** Baseline characteristics medication use of 10530 Belgian workers by gender.

Variables	Male	Female	*p*-Value
Being on medication for MSD			
No	5325 (98%)	4592 (96%)	<0.0001
Yes	136 (2%)	177 (4%)	
Being on medication for CVD			
No	5371 (93%)	4426 (93%)	0.418
Yes	390 (7%)	343 (7%)	
Being on medication for NPD			
No	5625 (98%)	4530 (95%)	<0.0001
Yes	136 (2%)	239 (5%)	
Chi-squared tests were used			

**Table 3 ijerph-17-07477-t003:** Logistic regression model for being on medication for MSD by gender.

Variables		Male		Female	
		Coeff (95%CI)	*p*-Value	Coeff (95%CI)	*p*-Value
Intercept		−4.651 (−5.246 to −4.063)	0.000	−4.231 (−4.717 to −3.672)	0.000
BMI		0.031 (0.012 to 0.047)	0.001	0.024 (0.008 to 0.039)	0.001
Physical activity	Yes vs. No	−0.155 (−0.286 to −0.045)	0.008		
Age		0.012 (0.001 to 0.021)	0.006	0.025 (0.016 to 0.033)	0.000
Noise 87 dB	Yes vs. No	0.204 (0.040 to 0.337)	0.006		
Physical load	Yes vs. No	0.225 (0.053 to 0.391)	0.011		
Manual handling with loads	Yes vs. No	0.190 (0.030 to 0.351)	0.013	−0.227 (−0.379 to −0.054)	0.013
Manual lifting, holding, carrying	Yes vs. No	−1.094 (−1.505 to −0.943)	0.045		
Manual repetitive tasks	Yes vs. No	0.866 (0.330 to 1.228)	0.002	0.260 (0.082 to 0.410)	0.010
Handling static loads	Yes vs. No			0.486 (0.201 to 0.750)	0.000
Shift work with task-specific risk	Yes vs. No	0.532 (0.256 to 0.751)	0.000		
log.psi1		4.639 (4.490 to 4.803)	0.000	4.380 (4.229 to 4.516)	0.000

**Table 4 ijerph-17-07477-t004:** Logistic regression model for being on medication for CVD by gender.

Variables		Male		Female	
		Coeff (95%CI)	*p*-Value	Coeff (95%CI)	*p*-Value
Intercept		−6.483 (−6.949 to −6.038)	0.000	−6.517 (−6.932 to −6.091)	0.000
BMI		0.049 (0.035 to 0.064)	0.000	0.053 (0.044 to 0.062)	0.000
Blood pressure	Elevated vs. Normal	−0.019 (−0.061 to 0.018)	0.465	−0.013 (−0.051 to 0.022)	0.570
	High Blood vs. Normal	−0.032 (−0.069 to −0.002)	0.159	0.009 (−0.023 to 0.045)	0.628
Smoking habits	Yes vs. No	−0.211 (−0.299 to −0.123)	0.000		
Age		0.076 (0.068 to 0.084)	0.000	0.076 (0.067 to 0.086)	0.000
Change in employment (Yes vs. No)		0.660 (0.389 to 1.023)	0.000	0.622 (0.294 to 0.918)	0.000
Stress at work	Yes vs. No	0.207 (0.024 to 0.388)	0.033		
Noise 80 dB	Yes vs. No	0.120 (−0.036 to 0.241)	0.043		
Physical load	Yes vs. No			0.101 (0.023 to 0.189)	0.039
Manual handling with loads	Yes vs. No	0.149 (0.080 to 0.230)	0.000		
Static load	Yes vs. No	0.273 (0.134 to 0.415)	0.000		
Shift work without task-specific risk	Yes vs. No	0.168 (0.085 to 0.251)	0.000		
Night work without task-specific risk	Yes vs. No	0.166 (0.035 to 0.312)	0.012		
Night work with task-specific risk	Yes vs. No	0.167 (−0.041 to 0.307)	0.049		
Psychosocial load	Yes vs. No	0.214 (0.073 to 0.326)	0.001	0.193 (0.087 to 0.278)	0.000
log.psi1		7.128 (7.004 to 7.318)	0.000	7.039 (6.902 to 7.206)	0.000

Elevated blood pressure: 120 ≤ Systolic ≤ 129 & Diastolic < 80; High blood pressure: Systolic ≥ 140 & Diastolic ≥ 90.

**Table 5 ijerph-17-07477-t005:** Logistic regression model for being on medication for NPD by gender.

Variables		Male		Female	
		Coeff (95%CI)	*p*-Value	Coeff (95%CI)	*p*-Value
Intercept		−3.055 (−3.849 to −2.391)	0.000	−4.080 (−4.800 to −3.355)	0.000
BMI		−0.019 (−0.041 to 0.004)	0.058		
Sport	Yes vs. No	−0.179 (−0.302 to −0.058)	0.002	−0.144 (−0.242 to −0.040)	0.002
Smoking habits	Yes vs. No			0.361 (0.189 to 0.539)	0.000
Age		0.020 (0.009 to 0.032)	0.000	0.017 (0.007 to 0.028)	0.000
Skill level	2 vs. 1	−0.532 (−0.855 to −0.113)	0.001	−0.251 (−0.506 to −0.008)	0.030
	3 or 4 vs. 1	−0.466 (−0.915 to −0.005)	0.013	−0.115 (−0.356 to 0.151)	0.298
Sector	Healthcare vs. Education			0.789 (0.220 to 1.461)	0.003
	Government vs. Education			0.858 (0.190 to 1.626)	0.003
	Food vs. Education			0.849 (−0.952 to 2.014)	0.081
	Distributive trade vs. Education			0.116 (−0.732 to 0.917)	0.781
	Manufacturing vs. Education			0.679 (−0.045 to 1.363)	0.050
	Services vs. Education			0.271 (−2.353 to 1.425)	0.609
	Transport vs. Education			0.723 (−0.892 to 1.658)	0.354
	Other vs. Education			0.588 (−7.639 to 2.052)	0.212
Noise 80 dB	Yes vs. No	0.458 (0.185 to 0.709)	0.001		
Physical load	Yes vs. No			0.227 (0.034 to 0.396)	0.014
Night work without task-specific risk	Yes vs. No	0.296 (0.012 to 0.583)	0.033		
Night work with task-specific risk	Yes vs. No			0.443 (0.012 to 0.787)	0.008
log.psi1		5.722 (5.530 to 5.916)	0.000	5.039 (0.012 to 0.787)	0.000

Elevated blood pressure: 120 ≤ Systolic ≤ 129 & Diastolic < 80; High blood pressure: Systolic ≥ 140 & Diastolic ≥ 90.

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
