# Peer review of "The Impact of a Change in Employment on Three Work-Related Diseases: A Retrospective Longitudinal Study of 10,530 Belgian Employees"

_ijerph, 2020, doi:10.3390/ijerph17207477_

Round 1

Reviewer 1 Report

Dear Authors,

many thanks for your article.

first of all I believe that a full revision of the English language of the paper is mandatory before publication.

Moreover, several major revisions are needed, as listed below:

Abstract

Line 20: please add “factors” after “The following”.

Lines 24-25: please, reformulate the sentence. It is not clear the meaning.

Lines 29-30: please change “with the risk of the medication used for NPD.” with “with the use of medications for NPD”

Lines 31-35: please, re-write the conclusions in a more clear and efficient way, reporting the main message of the study.

Introduction

Line 40: please change “anecdotally related” with reported. Please change 12 with twelve.

Line 44: please reformulate “to escape an incompetent boss”.

Line 46: please change “poorer” with “poor”

Lines 60-63: please, try to better explain the concepts reported. As the paper addresses the topic of the occupational medicine, it is not acceptable a too generic and not precise definition of the relations between work and health. In occupational medicine the physical risk is a very precise concept. In general, work-related risks that may impact workers’ health can be classified as physical, chemical, biological and related to work-organization (including work-stress).

Lines 70-72: reported information from the studies cited too generic, not possible to understand what is the purpose of the authors in citing them.

Lines 72-79: please. Re-write trying to explain what are the concepts reported in the studies cited that you are interested in. You should think that the readers may not have a full knowledge of all the studies you are citing. I am also not in favor on the way used of reporting e.g. “neither Cherry nor Metcalfe: the ones cited did not authored the papers alone. “Theorell reached the same conclusion… multiple changes in employment … a Swedish sample”: what kind of changes? what kind of sample? “physiological condition and the number of changes in employment or cardiac health” seem not the same as “myocardial infarction”.

Material and Methods

Ethical committee approval?

Lines 102-3: blood pressure and BMI are not risk factors

Lines 120-6: it is not clear how the occupational risks for the different jobs have been desumed and classified

Results

Line 155-6: please check this information. It is a very very high exposure level (i.e. it is actually the occupational limit) and 51% seems a very very jigh percentage of workers exposed to this level. Especially when compared to the data in Table 1, showing that only the 5-1& of subjects are exposed to 85dB, and 2-1% to 80 dB. This kind of exposure levels seem typical of many many years ago or of very specific occupational sectors (e.g. construction and metal work, but only some job tasks, not all the workers employed).

Table 1: all the occupational risks and their classification should be previously defined.

What is the difference between elevated and high blood pressure?

Lines 176-77: please avoid the use of words inferring causal relations. You only evaluated associations, not effects.

Lines 177: exposure to noise levels in excess of 87dB. Not coherent with previous text and table. Is it “of” or “in excess”? What is the number of workers above 87 dB? (note that this level should not be allowed according to current European legislation https://osha.europa.eu/it/legislation/directives/82).

Lines 178 vs 180: what is the difference between “manual handling of loads” and “manual lifting”? In my opinion one can consider that “manual lifting” is a subgroup of “manual handling of loads”. It can sound strange that one is risky and the other protective. Please clarify, and again better define how occupational risks were evaluated.

Table 3: again, differences with the previous text and new occupational risks introoduced here for the first time: Previously “manual handling”, here in the table “manual dealing”. In the table: noise 87 dB: equal? Above? And what about people exposed to 86 dB? (wich is for sure a very risky level as well).

Lines 187 and similar: when reporting ORs and CIs, if needed, please use also [ ] , not only ( )

Line 194: please double check the confidence intervals as it sounds strange to me that the Cis are not wide, considering that only the 3% of the population is classified as “psychosocial load yes”. By the way, what is psychosocial load?

Table 4: again, new risks here “Static load”. What is a “Shift work without particular risk”: I don’t think there are examples of shift jobs without occupational health hazards.

Lines 203-5_ not clear. Please, re-write.

Why for NPD you use the 80 dB noise cut off while previously the 87?

Discussion

The sensation is that only a small part of the results is discussed. Accordingly, why presenting all that results? I suggest to reduce the number of occupational risks considered.

Conclusions

“To our knowledge, this is the first study of its kind”: please delete.

Conclusions should be re-written according to a better definition of the occupational risks considered.

Funding:: difficult to believe that a so big work was not granted with research funds.

References

Please follow the IJERPH indications for reporting references.

Author Response

Dear Authors, many thanks for your article. first of all I believe that a full revision of the English language of the paper is mandatory before publication. Moreover, several major revisions are needed, as listed below:

General reply to Reviewer 1:

We wish to thank you for your constructive comments in this round of review. Your comments provided valuable insights to refine its contents and analysis. In this document, we try to address the issues raised as best as possible. We regret you have found problems with the English. The paper had been carefully revised by a native English speaker that had done also a professional language editing service of the first version of our manuscript to improve grammar and readability.

Comment 1:

Abstract

Line 20: please add “factors” after “The following”.

Lines 24-25: please, reformulate the sentence. It is not clear the meaning.

Lines 29-30: please change “with the risk of the medication used for NPD.” with “with the use of medications for NPD”

Lines 31-35: please, re-write the conclusions in a more clear and efficient way, reporting the main message of the study.

Reply to comment 1:

We have made the revision accordingly. We added the word “factors” after “The following”, changed “with the risk of the medication used for NPD.” with “with the use of medications for NPD”. Moreover, we reformulated the sentence in the conclusion as follow: “Change in employment and psychosocial load were found as two important risk factors for being on medication for CVD. Dealing with loads, doing shift work and being daily exposed to the noise of 87dB correlated with being on medication for MSD. Dealing with physical loads, doing night work and being exposed to the noise of 80dB were risk factors for being on medication for NPD. While doing physical activity and reporting higher skill levels were found to be protective factors for NPD.”.

Comment 2:

Introduction

Line 40: please change “anecdotally related” with reported. Please change 12 with twelve.

Line 44: please reformulate “to escape an incompetent boss”. <- incompatibility problems with the boss

Line 46: please change “poorer” with “poor”

Reply to comment 2:

We did all the above suggestion proposed by the reviewer and we reformulated “to escape an incompetent boss” as “incompatibility problems with the boss”.

Comment 3:

Lines 60-63: please, try to better explain the concepts reported. As the paper addresses the topic of the occupational medicine, it is not acceptable a too generic and not precise definition of the relations between work and health. In occupational medicine the physical risk is a very precise concept. In general, work-related risks that may impact workers’ health can be classified as physical, chemical, biological and related to work-organization (including work-stress)

Reply to comment 3:

We would like to thank the review for this suggestion. We added in the text the precise definition of work-related risks. “Work-related risks that may impact workers’ health can be grouped as biological, physical, ergonomic, chemical, and psycho-social. National Institute for Occupational Safety and Health (NIOSH) has reported that there exist 29 kinds of physical, 25 kinds of chemical, 24 kinds of biological, 6 kinds of ergonomic, and 10 kinds of psychosocial hazards (as work-stress).”

Comment 4:

Lines 70-72: reported information from the studies cited too generic, not possible to understand what is the purpose of the authors in citing them

Reply to comment 4:

Thank you for the suggestion. Our purpose is to highlight that the literature on this phenomenon is contradictory or shows inconsistent results. We modified the sentence as follow: “In a sample of Swedish male workers, poor mental health has been found to be weakly associated with the frequency of a change in employment, adjusted for socio-economic status and mediated by general mental health [Isaksson, K.A longitudinal study of the relationship between frequent job change and psychological well‐being. Journal of Occupational Psychology, 1990, 63(4), 297-308].”

Comment 5:

Lines 72-79: please. Re-write trying to explain what are the concepts reported in the studies cited that you are interested in. You should think that the readers may not have a full knowledge of all the studies you are citing. I am also not in favor on the way used of reporting e.g. “neither Cherry nor Metcalfe: the ones cited did not authored the papers alone. “Theorell reached the same conclusion… multiple changes in employment … a Swedish sample”: what kind of changes? what kind of sample? “physiological condition and the number of changes in employment or cardiac health” seem not the same as “myocardial infarction”.

Reply to comment 5:

We revised the entire paragraph in the following way: “In a sample of Swedish male workers, poor mental health has been found to be weakly associated with the frequency of a change in employment, adjusted for socio-economic status and mediated by general mental health [Isaksson, K.A longitudinal study of the relationship between frequent job change and psychological well‐being. Journal of Occupational Psychology, 1990, 63(4), 297-308]. Another study, in a sample of male employees, found the highest mortality risk in association with a series of changes among unrelated jobs. This sample was retrieved from the Stanford-Terman longitudinal study, an archive containing detailed work and life histories on approximately 1,500 men and women. Education, occupation, physical health, anxiety and depression were found as significant risk factors [Pavalko E.; Elder G. Jr.; Clipp E. Worklives and Longevity: Insights from a Life Course Perspective. Journal of Health and Social Behavior. 1993;34(4):363–380]. In a sample of young Dutch subjects, voluntary changes in employment was associated with better health condition. The study included work perceptions (organizational, departmental and task-related as explanatory variables) and job satisfaction, organizational commitment, intention to leave, absenteeism and tardiness as work outcomes [van der Velde, M.E.; Feij, J.A. Change of work perceptions and work outcomes as a result of voluntary and involuntary job change. Journal of Occupational and Organizational Psychology, 1995, 68(4), 273-290]. Finally, in a cohort of members of the National Survey of Health and Development, a longitudinal study of people born throughout Britain, it was found that early job changing is an indicator of later psychiatric problems [Cherry, N. Persistent job changing—Is it a problem?. Journal of occupational psychology, 1976, 49(4), 203-221]. Conversely, other studies did not find any evidence between frequent job changes in employment and the worker’s health status. This result was found for cardiac health in a sample of Scottish employees [Metcalfe, C.; Smith, G.D.; Sterne, J.A.; Heslop, P.; Macleod, J.; Hart, C. Frequent job change and associated health. Social science & medicine, 2003, 56(1), 1-15] and myocardial infarction in a case-control study in Sweden [Theorell, T. Life events before and after the onset of a premature myocardial infarction. 1974].

Material and Methods

Ethical committee approval?

Reply:

Thank you for the suggestion. We added the following sentence in “Population and study design” section: “The study protocol received the ethical approval from the Ethical Committee of Azienda Ospedaliera Policlinico “Paolo Giaccone” of Palermo (Nr. 8/2020, dated 23 September, 2020)”.

Lines 102-3: blood pressure and BMI are not risk factors

Reply:

We change the sentence as following: “physical and behavioural characteristics (such as high blood pressure, overweight and obesity measured through BMI, smoking habits)”.

Lines 120-6: it is not clear how the occupational risks for the different jobs have been desumed and classified

Reply to comment:

Thank you for the comment. We added the following sentence, in “Data collection and variables” section, to clarify this concept: “According to Belgian legislation, occupational physicians perform an individual risk assessment based on information received from the employer and information given by each employee during the medical examination. The occupational physician encodes risk factors using an electronic record system, given a list of risk codes that are briefly described using an overview list. This list was designed to leave space to the interpretation of the physician, based on his/her own experience and based on the needs of every specific employer or sector at hand. From this list of over 400 risk codes, a small subset was selected by two occupational physicians in an independent way, until consensus was reached. (See Table 1 in supplementary materials).”

Comment:

Results

Line 155-6: please check this information. It is a very very high exposure level (i.e. it is actually the occupational limit) and 51% seems a very very jigh percentage of workers exposed to this level. Especially when compared to the data in Table 1, showing that only the 5-1& of subjects are exposed to 85dB, and 2-1% to 80 dB. This kind of exposure levels seem typical of many many years ago or of very specific occupational sectors (e.g. construction and metal work, but only some job tasks, not all the workers employed).

Reply to comment:

This is a longitudinal retrospective study that includes 10,530 employees followed for at least seven measurement time points between 1993 and 2019. The measurement in Table 1 are at the baseline and for this reason, the included subjects could present old exposure levels linked to specific occupational sectors. We added in the results the following sentence: “Considering that the study is a retrospective (historical) cohort percentage of people exposed to daily noise levels of 87dB could appear oversized”.

Comment:

Table 1: all the occupational risks and their classification should be previously defined.

Reply to comment:

Thank you for the important suggestion. The procedure for assigning risks to employees has been added to “Data collection and variables” section and can be found in Table 1 in supplementary materials it is possible to find the occupational risks’ list.

Comment:

What is the difference between elevated and high blood pressure?

Reply to comment:

Regards blood pressure, we have used the American Heart Association classification.  We added in the text the following sentence: “The blood pressure was categorized according to the American Heart Association classification. This classification considers both systolic and diastolic blood pressure. The variable was categorized as “Normal”, “Elevated”, and “High”. Elevated for systolic blood pressure between 120-129 and diastolic blood pressure less than 80 and High blood pressure for a systolic pressure more or equal than 140 and a diastolic pressure more or equal than 90.”

Comment:

Lines 176-77: please avoid the use of words inferring causal relations. You only evaluated associations, not effects.

Reply to comment:

We changed influence with “be associated to”

Comment:

Lines 177: exposure to noise levels in excess of 87dB. Not coherent with previous text and table. Is it “of” or “in excess”? What is the number of workers above 87 dB? (note that this level should not be allowed according to current European legislation https://osha.europa.eu/it/legislation/directives/82).

Reply to comment:

We are extremely grateful to the Reviewer since it was a typo mistake. We have corrected in the entire paper removing the words “in excess”.

Comment:

Lines 178 vs 180: what is the difference between “manual handling of loads” and “manual lifting”? In my opinion one can consider that “manual lifting” is a subgroup of “manual handling of loads”. It can sound strange that one is risky and the other protective. Please clarify, and again better define how occupational risks were evaluated.

Reply to comment:

There might be confusion about the difference between both risks: ‘Manual handling of loads’ is a most general one; ‘manual lifting’ is more specific. As we performed our analysis at a sub-group and group level, we think that this result could be the effect of the so-called Simpson’s paradox. See also answer above: the procedure for assigning risks to employees has been added in “Data collection and variables” section.

Comment:

Table 3: again, differences with the previous text and new occupational risks introoduced here for the first time: Previously “manual handling”, here in the table “manual dealing”. In the table: noise 87 dB: equal? Above? And what about people exposed to 86 dB? (wich is for sure a very risky level as well).

Reply to comment:

Thank you for the valuable suggestion. This was a typo and has been changed in the text. We adopted the sentence “manual handling” on the entire text. The codification is 80<dB≤85, 85<dB≤87, and the last one is >87dB.

Comment:

Lines 187 and similar: when reporting ORs and CIs, if needed, please use also [ ] , not only ( )

Reply to comment:

We are grateful for this suggestion. We used both the square and the round parenthesis.

Comment:

Line 194: please double check the confidence intervals as it sounds strange to me that the Cis are not wide, considering that only the 3% of the population is classified as “psychosocial load yes”. By the way, what is psychosocial load?

Reply to comment:

The confidence interval for a frequency is as less wide as more the percentage is around the extreme 0% or 100%. Conversely, the confidence interval is as wider as bigger the sample size and for percentage equal to 50%, due to presence of maximum variability. We added the following sentence in the Data collection and variables section: “Psychosocial load is related to quantitative job demands, conflicting job demands, skill discretion, decision authority, supervisor support, co-worker support, and job security”

Table 4: again, new risks here “Static load”. What is a “Shift work without particular risk”: I don’t think there are examples of shift jobs without occupational health hazards.

Agreed: this might be confusing. We’ve included a table containing all the risks included in the study in Table 1 in supplementary material (see also comment above). ‘Without particular risk’ was substituted by “task-specific risk”.

Comment:

Lines 203-5_ not clear. Please, re-write.

Reply to comment:

Thank you for the suggestion. We have rewrite the sentence as follow: “To be on medication for NPD was significantly associated with age (OR= 1.02 95%CI[1.01-1.03] for male, OR=1.01 95%CI[1.01-1.02] for female), negatively associated with physical activity (OR= 0.83 95%CI[0.74-0.94] for male, OR=0.86 95%CI[0.78-0.96] for female). Furthermore, males with skill level 3 or 4 were protected compared to skill level 1 (OR= 0.63 95%CI[0.40-0.99]) and both males and females with skill level 2 were protected compared to individuals with a skill level 1 (OR= 0.58 95%CI[0.42-0.89] for male, OR=0.77 95%CI[0.60-0.99] for female).”

Comment:

Why for NPD you use the 80 dB noise cut off while previously the 87?

Reply to comment:

In the tables representing multivariable analysis we prefer to include only significant variables. For this reason, only 80dB noise was reported in Table 4. We added in the statistical methods’ section the following sentence: “For the sake of simplicity, in the multivariable analysis only coefficients with significant p-values were reported.”

Comment:

Discussion

The sensation is that only a small part of the results is discussed. Accordingly, why presenting all that results? I suggest to reduce the number of occupational risks considered.

Reply to comment:

Thank you for the important suggestion. Since this reduction will imply a loss of important information, we decided to not reduce the number of occupational risks.

Regarding to CVD, we added the following sentence: “Our study found that physical load, manual handling with loads and static load are risk factors for CVD. This is in accordance with Hannerz and Holtermann’s study where employment in occupations that involve heavy lifting is a predictor for Ischemic Heart Disease [Hannerz, H.; Holtermann, A. Heavy lifting at work and risk of ischemic heart disease: protocol for a register-based prospective cohort study. JMIR Research Protocols, 2014, 3(3), e45].”

Regarding NPD we added the following sentences and the related references:

The presence of respiratory system complaints caused by smoking habits was found to be an important risk factor for a broad category of mental disease as anxiety, depression, and the combination of anxiety and depression [Ergün, D.; Ergün, R.; Ergan, B.; Kurt, Ö.K. Occupational risk factors and the relationship of smoking with anxiety and depression. Turkish thoracic journal, 2018, 19(2), 77].

Stansfeld et al. demonstrated that the type of occupation is an important risk factor for common mental disorder. It happens because “occupations may be typified by high levels of job demands, especially emotional demands and lack of job security. The reasons why occupations have low rates of common mental disorder are varied and may include high levels of job discretion, good job training and clearly defined job tasks” [Stansfeld, S.A.; Rasul, F.R.; Head, J.; Singleton, N. Occupation and mental health in a national UK survey. Social psychiatry and psychiatric epidemiology, 2011, 46(2), 101-110.].

Mental health is demonstrated to be negatively correlated with the constant high physical and psychosocial work demands [Hiesinger, K.; Tophoven, S. Job requirement level, work demands, and health: a prospective study among older workers. International archives of occupational and environmental health, 2019, 92(8), 1139-1149].

Comment:

Conclusions

“To our knowledge, this is the first study of its kind”: please delete.

Reply to comment:

We are grateful to the reviewer for this suggestion. We rephrased the sentence in the following way: “the aim of this study was to explore the effect of changes in employment on the health of employees, while considering a series of risk factors associated with work-related diseases.”

Comment:

Conclusions should be re-written according to a better definition of the occupational risks considered.

Reply to comment:

We have modified the conclusion adding the following sentence: “As a consequence, can be recommended to track the employer's working life by recording his/her job changes. In fact, the worsening of the worker’s health reflects in lower productivity, augmented cost for the employer and affects public health. Our study showed an increased risk of cardiovascular disease in Belgian workers who experience a job change. Therefore, it is advisable, especially in the transition phases between one job and another, that the worker’s health status is monitored by the general practitioner and the occupational physician.”

Funding:: difficult to believe that a so big work was not granted with research funds.

Comment:

The study data were obtained from the private largest central repository of data on Belgian employees, the IDEWE data warehouse. It is an administrative database including data from the annual health checks of Belgian employees. None of the authors of this paper received any research funds for this work.  However, the following co-authors Martijn Schouteden, Jan Boon and Lode Godderis collaborate with IDEWE. We included a sentence in the conflict of interest’s section.

References

Please follow the IJERPH indications for reporting references.

Reply: We have made the revision accordingly.

Reviewer 2 Report

I realize that a great work and time has been devoted to this paper. It is about the impact of a change in employment on three work-related diseases, on cardiovascular (CVD), musculoskeletal (MSD) and neuropsychological (NPD) diseases in Belgian subjects in a 7-year follow-up study period with a sample size of 10,530 participants.  This is a topic of great significance to emotional wellbeing of employees that can affect the quality of life, and therefore, be a public health problem. So I appreciate authors examining this topic.

The paper has a lot of strengths but I think that some changes should be recommended.

 Abstract:

Readers should be able to read the abstract in isolation and understand what you have done, and its implications. The background is not only the aim of the study.

It is not necessary to cite “the authors of this paper hope…” because it detract from the findings and conclusions. Please, avoid it.

Please, avoid using abbreviations in the abstract.

Introduction:

The introduction is very scarce, severe lacking of references and does not justify why it has conducted this study.

Methodology:

The authors must specify the type of sampling, the response rates, if the participation was volunteer and anonymous, the inclusion criteria, loss to follow-up, etc.

It is unclear why do you have selected nurses and physicians and the results do not compare both professional groups.

Could you specify how were conducted the questionnaires (e.g. e-mail, postal mail, at the workplace,)? And please, specify how long it takes to the employees performing all surveys.

It is unclear if the instruments used for demographic characteristics and job conditions were ad-hoc. Please, you have to specify this in the text.

I'm not sure if validated questionnaires have been used to measure stress at work, medication use and other variables studied. So could you specify in the text the Cronbach’s Alpha Coefficient for each of questionnaires if the authors used them?

Results:

 The results are good and exhaustive.

Discussion:

There are no limitations of this study and the discussion is severely lacking.

Conclusions:

I would suggest to the authors to avoid the word “we/our”, writing in impersonal mode as scientific style.

Your study has a lot of implications for nurses, physicians, hospital managements and even the patients. Please, can you elaborate some implications for practice and strategies to improve this situation? 

I hope that these recommendations do not discourage the authors and I want to recommend the authors to continue working on this paper, because it could be publishable if mayor changes are made.

Author Response

I realize that a great work and time has been devoted to this paper. It is about the impact of a change in employment on three work-related diseases, on cardiovascular (CVD), musculoskeletal (MSD) and neuropsychological (NPD) diseases in Belgian subjects in a 7-year follow-up study period with a sample size of 10,530 participants.  This is a topic of great significance to emotional wellbeing of employees that can affect the quality of life, and therefore, be a public health problem. So I appreciate authors examining this topic. The paper has a lot of strengths but I think that some changes should be recommended.

General reply to Reviewer 2:

We wish to thank you for your constructive comments in this round of review and the support. Your comments provided valuable insights to refine its contents and analysis. In this document, we try to address the issues raised as best as possible.

Comment 1:

Abstract:

Readers should be able to read the abstract in isolation and understand what you have done, and its implications. The background is not only the aim of the study.

Reply to comment 1: We are grateful to the reviewer for this suggestion. The abstract’s background was changed as follow: “The literature investigating to what extent does a change in employment contribute to good health is contradictory or shows inconsistent results. The aim of this study was to investigate whether there exists an association between change in employment and cardiovascular, musculoskeletal and neuropsychological diseases in a sample of 10,530 Belgian workers in a 7-year follow-up study period.”

Comment 2:

It is not necessary to cite “the authors of this paper hope…” because it detract from the findings and conclusions. Please, avoid it.

Reply to comment 2: We have made the revision deleting the sentence “the authors of this paper hope…”.

Comment 3:

Please, avoid using abbreviations in the abstract.

Reply to comment 3: We have made the revision accordingly and then the abbreviations were avoided.

Comment 4:

Introduction:

The introduction is very scarce, severe lacking of references and does not justify why it has conducted this study.

Reply to comment 4:

Thank you for your valuable suggestion. We completely modified the introduction adding more explanation on the cited studies and we included several references. The reason why we conducted this study was because there is not a lot of literature present at the time of study; and the existing publications sometimes reported contradictory or inconsistent results. So, it seemed worth the effort to investigate further. For this reason, we have modified the last part of the introduction in the following way: “Given the controversial nature of these wide-ranging outcome variables, the authors of this study contend that the possible association between frequent changes in employment and health merits further attention. The literature investigating to what extent does a change in employment contribute to good health is contradictory or shows inconsistent results. For, this reason the aim of this study was to detect whether there exists an association between the impact of a change in employment on cardiovascular (CVD), musculoskeletal (MSD) and neuropsychological (NPD) diseases in worker’s employed in different sectors in a 7-year follow-up study period. The association between changes in employment and being on medication for the aforementioned work-related diseases will be scrutinised, as proxies for the three health problems, while controlling for work-related risks.”

Comment 5:

Methodology:

The authors must specify the type of sampling, the response rates, if the participation was volunteer and anonymous, the inclusion criteria, loss to follow-up, etc. It is unclear why do you have selected nurses and physicians and the results do not compare both professional groups. 

Reply to comment 5:

Data inside the data warehouse had been gathered and updated annually by occupational nurses and physicians. IDEWE includes data on Belgian employees that belong to several sectors. We have deleted the sentence “which has been garnered by occupational nurses and physicians” that could generate confusion.

Comment 6:

Could you specify how were conducted the questionnaires (e.g. e-mail, postal mail, at the workplace,)? And please, specify how long it takes to the employees performing all surveys.

Reply to comment 6:

No questionnaires were used. We modified the sentence in “Population and study design” section as follows: “In addition to medical data, personal and work characteristics are also registered and encoded during medical examination, using international or national classifications standards in electronic records.”

Comment 7:

It is unclear if the instruments used for demographic characteristics and job conditions were ad-hoc. Please, you have to specify this in the text.

Reply to comment 7:

Data were extracted from the administrative database obtained by the annual health checks of Belgian employees. References cited in the text provide a lot of information about our medical examination process [Godderis, L.; Mylle, G.; Coene, M.; Verbeek, C.; Viaene, B.; Bulterys, S.; Schouteden, M. Data warehouse for detection of occupational diseases in OHS data. Occupational Medicine, 2015, 65(8), 651-658.] and [Godderis, L.; Johannik, K.; Mylle, G.; Bulterys, S.; Moens, G. Epidemiological and performance indicators for occupational health services: a feasibility study in Belgium. BMC health services research, 2014, 14(1), 410.].

Comment 8:

I'm not sure if validated questionnaires have been used to measure stress at work, medication use and other variables studied. So could you specify in the text the Cronbach’s Alpha Coefficient for each of questionnaires if the authors used them?

Reply to comment 8:

Thank you for your comment. No questionnaire was used to measure stress; the doctor makes the judgement and assigns the risk codes. We added the following sentence, in “Data collection and variables” section, to clarify this concept: “According to Belgian legislation, occupational physicians perform an individual risk assessment based on information received from the employer and information given by each employee during the medical examination. The occupational physician encodes risk factors using an electronic record system, given a list of risk codes that are briefly described using an overview list. This list was designed to leave space to the interpretation of the physician, based on his/her own experience and based on the needs of every specific employer or sector at hand. From this list of over 400 risk codes, a small subset was selected by two occupational physicians in an independent way, until consensus was reached. (See Table 1 in supplementary materials).”

Comment 9:

Results:

 The results are good and exhaustive.

Reply to comment 9:

We would like to thank the reviewer.

Comment 10:

Discussion:

There are no limitations of this study and the discussion is severely lacking.

Reply to comment 10:

We have added several sentences in the discussion. In particular, Regarding to CVD, we added the following sentence: “Our study found that physical load, manual handling with loads and static load are risk factors for CVD. This is in accordance with Hannerz and Holtermann’s study where employment in occupations that involve heavy lifting is a predictor for Ischemic Heart Disease [Hannerz, H.; Holtermann, A. Heavy lifting at work and risk of ischemic heart disease: protocol for a register-based prospective cohort study. JMIR Research Protocols, 2014, 3(3), e45].”

Regarding NPD we added the following sentences and the related references:

Regarding to CVD, we added the following sentence: “Our study found that physical load, manual handling with loads and static load are risk factors for CVD. This is in accordance with Hannerz and Holtermann’s study where employment in occupations that involve heavy lifting is a predictor for Ischemic Heart Disease [Hannerz, H.; Holtermann, A. Heavy lifting at work and risk of ischemic heart disease: protocol for a register-based prospective cohort study. JMIR Research Protocols, 2014, 3(3), e45].”

Regarding NPD we added the following sentences and the related references:

The presence of respiratory system complaints caused by smoking habits was found to be an important risk factor for a broad category of mental disease as anxiety, depression, and the combination of anxiety and depression [Ergün, D.; Ergün, R.; Ergan, B.; Kurt, Ö.K. Occupational risk factors and the relationship of smoking with anxiety and depression. Turkish thoracic journal, 2018, 19(2), 77].

Stansfeld et al. demonstrated that the type of occupation is an important risk factor for common mental disorder. It happens because “occupations may be typified by high levels of job demands, especially emotional demands and lack of job security. The reasons why occupations have low rates of common mental disorder are varied and may include high levels of job discretion, good job training and clearly defined job tasks” [Stansfeld, S.A.; Rasul, F.R.; Head, J.; Singleton, N. Occupation and mental health in a national UK survey. Social psychiatry and psychiatric epidemiology, 2011, 46(2), 101-110.].

Mental health is demonstrated to be negatively correlated with the constant high physical and psychosocial work demands [Hiesinger, K.; Tophoven, S. Job requirement level, work demands, and health: a prospective study among older workers. International archives of occupational and environmental health, 2019, 92(8), 1139-1149].

Moreover, we added the section strength and limitations: “The main strength of the paper is the availability of a 7-years follow-up study period in order to investigate the association between job changes and three different diseases, while considering a wealth of work-related risk factors. Conversely, there are a few study limitations. First of all, the drop-out of employees that leave their job or change it, with the effect to be lost to follow up, because they are no longer enrolled in the same OSH provider (IDEWE). Moreover, the specific causes of job changes are not considered, so that it cannot be excluded the occurrence of some confounding in the analysis. Other important potential risk factors such as diet, work satisfaction, and sickness absence, family life, supportive relationships with colleagues, economic security, and access to social support were not measured. Self-reported information on smoking habits was potentially underreported and some risks as burnout, conflicts with customers, safety risk and hindrance were excluded to avoid sparseness in the models, since they presented very few observations.”

Comment 11:

Conclusions:

I would suggest to the authors to avoid the word “we/our”, writing in impersonal mode as scientific style.

Reply to comment 11: We have to thank the reviewer for its valuable suggestion. We have made the revision accordingly.

Comment 12:

Your study has a lot of implications for nurses, physicians, hospital managements and even the patients. Please, can you elaborate some implications for practice and strategies to improve this situation?

Reply to comment 12:

We added the following sentence in the conclusion’s section: “As a consequence, can be recommended to track the employer's working life by recording his/her job changes. In fact, the worsening of the worker’s health reflects in lower productivity, augmented cost for the employer and affects public health. Our study showed an increased risk of cardiovascular disease in Belgian workers who experience a job change. Therefore, it is advisable, especially in the transition phases between one job and another, that the worker’s health status is monitored by the general practitioner and the occupational physician.”

Comment 13:

I hope that these recommendations do not discourage the authors and I want to recommend the authors to continue working on this paper, because it could be publishable if mayor changes are made.

Reply to comment 13:

We would like to thank the reviewer for his/her valuable suggestions and for the support.

Round 2

Reviewer 1 Report

Dear Authors,

congratulations. 
I am definitely satisfied with your responsens and the revisions of the manuscript.

I believe that the article can be considered now ready for publication.

Best regards,

the Reviewer

Reviewer 2 Report

Dear Authors,

Congratulations on the work done and the changes made.

I think that the paper has improved.

I wish you luck!